# Development and Characterization of Liposomal Formulations Containing Phytosterols Extracted from Canola Oil Deodorizer Distillate along with Tocopherols as Food Additives

**DOI:** 10.3390/pharmaceutics11040185

**Published:** 2019-04-16

**Authors:** Asmita Poudel, George Gachumi, Kishor M. Wasan, Zafer Dallal Bashi, Anas El-Aneed, Ildiko Badea

**Affiliations:** Drug Design and Discovery Group, College of Pharmacy and Nutrition, University of Saskatchewan, 107 Wiggins Road, Saskatoon, SK S7N 5E5, Canada; asp170@mail.usask.ca (A.P.); george.gachumi@usask.ca (G.G.); kishor.wasan@usask.ca (K.M.W.); zafer.bashi@usask.ca (Z.D.B.)

**Keywords:** phytosterols, tocopherols, liposomes, canola oil deodorizer distillate, model orange juice

## Abstract

Phytosterols are plant sterols recommended as adjuvant therapy for hypercholesterolemia and tocopherols are well-established anti-oxidants. However, thermo-sensitivity, lipophilicity and formulation-dependent efficacy bring challenges in the development of functional foods, enriched with phytosterols and tocopherols. To address this, we developed liposomes containing brassicasterol, campesterol and β-sitosterol obtained from canola oil deodorizer distillate, along with alpha, gamma and delta tocopherol. Three approaches; thin film hydration-homogenization, thin film hydration-ultrasonication and Mozafari method were used for formulation. Validated liquid chromatographic tandem mass spectrometry (LC-MS/MS) was utilized to determine the entrapment efficiency of bioactives. Stability studies of liposomal formulations were conducted before and after pasteurization using high temperature short time (HTST) technique for a month. Vesicle size after homogenization and ultrasonication (<200 nm) was significantly lower than by Mozafari method (>200 nm). However, zeta potential (−9 to −14 mV) was comparable which was adequate for colloidal stability. Entrapment efficiencies were greater than 89% for all the phytosterols and tocopherols formulated by all three methods. Liposomes with optimum particle size and zeta potential were incorporated in model orange juice, showing adequate stability after pasteurization (72 °C for 15 s) for a month. Liposomes containing phytosterols obtained from canola waste along with tocopherols were developed and successfully applied as a food additive using model orange juice.

## 1. Introduction

Functional foods and nutraceuticals are increasing rapidly due to growing consumer preferences towards natural bioactives rather than synthetic drugs for disease prevention and treatment [1]. Phytosterols and tocopherols are such bioactives (plant metabolites) that have numerous health claims [2,3]. The primary health benefit of phytosterols is to lower low-density lipoprotein (LDL) cholesterol levels in plasma [2,4,5]. Due to this health claim, the National Cholesterol Education Program Adult Treatment Panel III (NCEP ATP III) has recommended phytosterols as adjuvant therapy to statins in hypercholesterolemia [6]. Phytosterols compete with cholesterol for their solubilization in bile salt micelles, hindering the absorption of cholesterol in blood [7,8]. Tocopherols, on the other hand, are free radical scavengers and natural anti-oxidants [9,10]. Due to their anti-oxidant properties, tocopherols are used in the treatment of age related macular degeneration [11], Alzheimer’s disease [12], glaucoma [3] and heart diseases [13].

Sources of phytosterols and tocopherols include oil seeds such as canola and sesame, as well as nuts [14,15,16]. Among these, canola is a major source for edible vegetable oils, and the most abundant oilseed crop in Canada [17]. It is a rich source of four phytosterols, namely beta-sitosterol, campesterol, stigmasterol and brassicasterol, and four tocopherols (alpha, beta, gamma and delta) [16,18]. Canola oil loses some of its valuable components during the refining process [16]. Significant amount of phytosterols and tocopherols are transferred to the waste stream, termed canola oil deodorizer distillate (CODD) [16] which offers an ideal source of these components.

However, formulation of these bioactives in functional foods has always been challenging due to their lipophilicity and light sensitivity [19]. In particular, degradation products of phytosterols, phytosterol oxidation products (POPs), are known to have some negative impact on human health [20,21]. Thus, the selection of suitable formulation approach is crucial during the development of functional food that contain these bioactives. Encapsulation techniques, such as spray drying, fluidized bed coating, microemulsification and liposomal entrapment are emerging in the food industry to address lipophilicity related challenges [22,23]. Unfortunately, most of these techniques have shortcomings such as usage of high temperature (can possibly degrade phytosterols and tocopherols) and the requirement of large quantities of emulsifiers and surfactants, which are deleterious to human health [23,24,25,26,27]. All of these shortcomings can be addressed by employing liposomal formulations that require low heat and low quantities of surfactants or emulsifiers [28].

Phytosterols in both the free and esterified forms have been used in the food industry [2,4]. Solubilization of esterified phytosterols in fat containing foods, like margarine [29,30], salad dressing [31] and yogurt [32] is prevalent in the food industry. However, this approach is not favorable to people who are on low fat diet [33]. To overcome this, various low fat or non-fat food matrices such as low fat milk [34,35], granola bars [36], orange juice [37] and non-fat beverages [38,39,40] are emerging as food products. However, for these type of food products, lipophilic phytosterols should be well formulated prior to their development into functional food. In addition to the choice of the food matrix, the biological efficacy should also be carefully considered.

Various clinical trials have shown that the efficacy of phytosterols depends on different parameters, such as solubility in the food matrix and the formulation [2,41]. Esterified phytosterols solubilized in fat/oil are driven favorably towards the bile salt micelles in the guts than the crystalline or the insolubilized forms [42,43]. Phytosterols ester containing food products such as milk, spread and yogurt have showed reduction in LDL-cholesterol by 7–12% at daily dose of 1.6–2 g relative to control research participants [44,45,46]. In contrast, some failed clinical trials are also prevalent [47,48]. For example, Ottestad et al. showed that phytosterol ester in the capsular formulation revealed no significant reduction of LDL cholesterol [47]. Similarly, Denke et al. showed no significance in cholesterol reduction by sitostanol capsule relative to control [48]. Unlike phytosterol capsule-based trials, lecithin-based free phytosterol formulations have shown to impart efficacy as high as 14.3% at a daily dose of only 1.9 g relative to control [39]. In sum, literature reports show that the efficacy of phytosterols depends greatly on the formulation approach, which provides insights regarding the possibility of further enhancing their efficacy by well formulating in suitable delivery systems.

The work of Shin at al. [38] and Spilburg at al. [39] provides a strong evidence that lecithin (phosphatiylcholine) can be effective carrier of phytosterols to increase cholesterol-lowering efficacy. Both of these studies used lecithin micelles to formulate sterol/stanol which have shown promising cholesterol-lowering efficacy [38,39]. Liposomes, which have same building blocks as micelles that is lecithin (i.e., phosphatidylcholine) but different architecture are another formulation strategy in which lecithin can be utilized, thus have potential of further enhancing its cholesterol-lowering efficacy. In addition, liposomes can prevent oxidation of thermo-sensitive bioactives and are biocompatible and biodegradable [49]. Further, co-formulation of tocopherols along with phytosterols can enhance oxidative stability of phytosterols [50].

Thus, in this work, with the aim of enhancing phytosterols’ oxidative stability and increasing its efficacy, we formulated phytosterols (obtained from CODD) and commercially available tocopherols into liposomes employing three different approaches, namely thin film hydration homogenization, thin film hydration ultra-sonication and Mozafari method. The liposomal formulation showing the highest entrapment efficiency, adequate size and zeta potential was incorporated into model orange juice (acidified solution). Thus, functional orange juice containing liposomal phytosterols and tocopherols was developed and its stability was assessed.

## 2. Materials

### Chemicals and Reagents

Phytosterols were extracted from CODD obtained from LDM foods (Yorkton, SK, Canada). Briefly, 5 g of CODD was saponified with 1 M potassium hydroxide in 95% ethanol for 1 h at 65 °C after which water was added and the mixture was chilled at 9.5 °C for 1 h. After the crystallization of phytosterols, vacuum filtration was performed and the residue was washed before being dried under high vacuum. Tocopherols, chloroform, ethyl acetate and potassium hydroxide were purchased from Sigma Aldrich (Oakville, ON, Canada), and phosphatidylcholine (PC) was purchased from Avanti Polar Lipids (Alabaster, AL, USA). Purified water was obtained from Millipore (Bedford, MA, USA).

## 3. Methods

### 3.1. Formulation of Liposomes

Three different formulation techniques namely thin film hydration homogenization; thin film hydration ultrasonication and Mozafari method were used for formulation in order to evaluate the formulation technique that can produce liposomes with optimum physicochemical properties for both oral delivery and industrial scale up.

#### 3.1.1. Method I. Thin Film Hydration–Homogenization

This method was adopted from Chung et al. [51] with some modifications. In brief, tocopherols (alpha, gamma and delta tocopherol), phytosterols mixture (brassicasterol, campesterol and beta-sitosterol) and PC were dissolved in 5 mL ethyl acetate (food grade) in 0.1:0.9:2, 0.1:0.9:3, 0.1:0.9:4 and 0.1:0.9:5 ratio of tocopherol: phytosterol: PC. Ethylacetate was evaporated using rotary evaporator at pressure of 90 mmHg. The thin lipid film containing bioactives and PC formed on the wall of the flask was lyophilized for 10 h to remove traces of ethylacetate and was hydrated with 20 mL of purified water for 3 h at 55°C with occasional vortexing in the presence of glass beads. The lipid dispersion was homogenized using recirculating high fluid pressure homogenizer (Microfluidics Corporation, Westwood, MA, USA) at 60 psi for 20 min. The prepared liposomes were left overnight at 4 °C prior to size analysis.

#### 3.1.2. Method II: Thin Film Hydration Ultrasonication

This method was adopted from Akbarzadeh et al. [52]. Similar to thin film hydration homogenization; tocopherols (alpha, gamma and delta tocopherol), phytosterols mixture (brassicasterol, campesterol and beta-sitosterol) were dissolved, along with PC in 5 mL ethyl acetate, in 0.1:0.9:2 ratio of tocopherols: phytosterols: PC. Ethylacetate was evaporated using rotary evaporator at pressure of 90 mmHg. Thin lipid film containing bioactives and PC was formed at the bottom of the flask. Lipid film was lyophilized for 10 h to remove traces of ethylacetate and was hydrated with 20 mL of purified water maintained at 55°C. Lipid dispersion was ultrasonicated using bath sonicator (ELMA Corp.,Singen, Germany) for 30 min maintained at 55 °C then was allowed to cool at room temperature. The prepared liposomes were left overnight at 4 °C prior to size analysis.

#### 3.1.3. Method III: Mozafari Method

This method was adopted from Colas et al. [53]. 50 mg of PC was hydrated with 20 mL of purified water for 1 h and was heated to 55 °C. Nine mg of the phytosterol mixture and 1 mg of the tocopherol mixture were heated with 3% *v*/*v* glycerol at 110 °C and 55 °C temperature, respectively for 15 min on a hot plate stirrer at approximately 1000 RPM (Corning Corporation, Midland, ON, Canada) and then was cooled down to 55 °C. PC dispersion, phytosterols and tocopherols were mixed together with stirring on a hot plate for 30 min at approximately 1000 RPM. The formed liposomes were cooled down to room temperature and kept overnight at 4 °C prior to size analysis.

### 3.2. Characterization of Particle Size, Size Distribution and Zeta Potential

Particle size and zeta potential measurement of the liposomes were performed using Zeta sizer, Nano ZS instrument, Malvern instruments Ltd. (Worcestershire, England). All measurements were conducted in triplicates at 25 °C and reported as mean ± SD.

### 3.3. Transmission Electron Microscopy (TEM) Analysis

TEM analysis was performed by negative staining. Briefly, a drop of liposomal sample was placed on copper- formvar coated TEM grid and was allowed to settle on grid surface for 1 min. Excess of the liquid was removed using absorbent tissue. Staining of grid was done using 0.5% phosphotungstic acid for 30 s and excess of stain is removed. Imaging was done using aHT 7700 TEM (Hitachi, Japan) at 80 kV.

### 3.4. LC-MS/MS Method Development and Validation

LC-MS/MS method was developed and was validated as per International Council for Harmonization of Technical Requirements for Pharmaceuticals for Human Use (ICH) guidance for bioanalytical method validation guideline [54]. Briefly, chromatographic separation of the analytes was carried out on an Agilent Acquity UPLC (Agilent Technologies, Mississauga, ON, Canada) with an Agilent Poroshell C18 column (2.1 mm × 150 mm, 5µm) protected by a guard column (2.1 mm × 4.7 mm, 2.7 µm) of the same packing material. The column temperature was set at 30 °C and the injection volume was 2.5 µL. An isocratic elution consisting of acetonitrile: methanol (99:1 *v*/*v*) with 0.1% acetic acid was used at a flow rate of 0.8 mL/min. The detection and quantification were performed using an API 6500 QTRAP^®^ quadruple-linear ion trap (QqQ-LIT) mass spectrometer equipped with an atmospheric pressure chemical ionization (APCI) source obtained from AB Sciex(Mississauga, ON, Canada). The instrument was operated in the positive ion mode and tandem mass spectrometric analysis (MS/MS) was employed using the following interface parameters: source temperature 380 °C, curtain gas 30 psi (gas), nebulizer current 2.5 µA, declustering potential 30 V and an ion source gas1 30 psi (gas) [55].

The parameters, selectivity, accuracy, precision, reproducibility, sensitivity, matrix effects, dilution integrity, stability were assessed [55]

### 3.5. Entrapment Efficiency (EE)

In order to determine entrapment efficiency, free and entrapped bioactives were separated using ultracentrifugation. Ultracentrifuge (Beckman coulter, Inc., Indianapolis, IN, USA) with rotor SW 60Ti was used for ultracentrifugation. Briefly, 5 mL of liposomes was ultracentrifuged at 30, 60, 90 and 120 min at constant RPM 32,000 (G-force of 138000). The sediment at each time were analyzed using a validated LC-MS/MS method to optimize ultracentrifugation parameters. The liposomes (present in sediment) separated by ultracentrifugation were lyophilized using freeze dryer for 24 h. Similar lyophilization process was employed with 5 mL of unseparated liposomes for 24 h. Dried unseparated and separated liposomes were dissolved in 2 mL of chloroform separately. Aliquot of each were spiked with internal standard and diluted with acetonitrile. Samples were injected in LC-MS along with freshly prepared calibration and quality control standards, as described [55]. The entrapment efficiency was calculated by measuring the ratio of entrapped bioactives in the formulation to the total bioactives present in the formulation and was determined using following equation:(1)$$\%\mathrm{Entrapment}\;\mathrm{efficiency}=\frac{E\;\mathrm{bioactives}}{T\;\mathrm{bioactives}}\times 100$$
where, E bioactives = Entrapped bioactives in liposomes (present in sediment of separated liposomes); T bioactives = Total bioactives in liposomes (present in unseparated liposomes).

### 3.6. Development of Functional Juice Using Model Orange Juice

In order to preserve particle size during freeze drying, sucrose was added to liposomes as a cryoprotectant by adopting the procedure of Shaikh et al. [56]. Briefly, 5% *w*/*v* of sucrose was added to liposomes of well-defined size and was vortexed. Lyophilization then was employed for 24 h.

Freeze dried liposomes were re-suspended in model orange juice which is an orange juice mimic at 3.2 pH. A mimic was used instead of real orange juice to enable particle size analysis without the interference of particulate components existing in the orange juice. In fact, the acidified solution is considered a model juice and was prepared by using acetic acid as per the protocol of Marsansco et al. [57]. This protocol can be applied for fruit juice with a pH less than 5.0, such as orange juice and pineapple juice. Finally, liposomes with optimum entrapment efficiency in a dried form were incorporated into the model orange juice by vortexing for 5 min.

### 3.7. Pasteurization

High temperature short time (HTST) pasteurization technique was employed as described [58]. HTST is a commonly used strategy for the pasteurization of juice [59]. The liposomes containing model orange juice was pasteurized at 72 °C for 15 s. Unpasteurized formulation was used as a control. Both pasteurized and unpasteurized model juice were stored at 4 °C for stability evaluation.

### 3.8. Chemical Stability Studies

Both pasteurized and non-pasteurized model juice containing liposomal bioactives were analyzed using LC-MS/MS to assess the degradation of bioactive upon exposure to pasteurization temperature. Briefly, 5 mL each of pasteurized and non-pasteurized liposomal model juice were lyophilized. Dried sample were dissolved in chloroform and were diluted with acetonitrile for LC-MS/MS analysis. The LC-MS/MS response was compared to obtain relative quantification data.

### 3.9. Physical Stability Studies

Physical stability evaluation was conducted at the interval of 7 days for a month. Particle size of pasteurized and non-pasteurized liposomes incorporated both in model orange juice were analyzed.

## 4. Statistical Analysis

The statistical analysis of the samples was conducted with SPSS statistical software version 24 (SPSS Inc., Chicago, IL, USA) using student *t*-test, and *p*-values < 0.05 were considered statistically significant. All data are reported as means ± standard deviations.

## 5. Results and Discussions

### 5.1. Physicochemical Characterization

Size is an important parameter to assess the stability, the biological fate and the efficacy of formulated bioactives [60,61]. Optimization of bioactive to lipid weight ratio (B/L ratio) was performed at ratio of 1:5, 1:4. 1:3, 1:2, 1:1 using liposomes prepared by thin film hydration- homogenization approach. An increase in particle size was observed at high B/L ratio (1:1) as shown in Figure 1. At 1:5 B/L ratio, the particle size was 149.53 nm; however, when B/L ratio increased to 1:1, the particle size increased to 258.31 nm (Figure 1). This observation is reported previously [62,63], in which incremental vesicle size was observed when increasing the cholesterol (a sterol) concentration. While 1:1 ratio is preferable from a commercial point of view (less PC required for formulation), smaller particle size (less than 200 nm) attained at 1:2 ratio, is optimum for liposomal stability. This optimum vesicle size (less than 200 nm) is consistent with several food-based liposomes [64,65,66]. Thus, 1:2 B/L ratio was selected for follow-up experiments. Same optimum B/L ratio was selected for thin-layer ultrasonication approach. However, in case of the Mozafari method, B/L higher than 1:5 led to the appearance of visible white precipitate. Loading techniques along with the preparation procedures are found to influence drug/lipid ratio of liposomes [67]. In both thin film hydration homogenization and thin film hydration ultrasonication, hydrated bioactives-PC film is subjected to cavitation and shearing forces unlike the Mozafari method where less intense magnetic stirring is used during the loading process. This might have led to differences in the B/L ratio of the mozafari method in comparison with ultrasonication and homogenization methods. In this way, 1:5 B/L as optimum ratio was selected for formulations prepared by the Mozafari method.

The comparison of the vesicle size using the different formulation strategies is presented in Table 1. Thin film hydration homogenization and thin film hydration ultrasonication were comparable, showing sizes at 186.33 ± 4.38 nm and 196.2 ± 16.1 nm, respectively. On the other hand, the size was significantly larger in the case of the Mozafari method (260 ± 22.98 nm). It is possible that the high shear force and cavitation involved in size reduction during the homogenization and ultrasonication methods is the reason for the obtained smaller vesicles [68]. The Mozafari method uses a less intense magnetic stirring [69], probably yielding larger particles. Polydispersibility index (PDI) shown in Table 1 was found to be in the range from 0.29 to 0.37, which shows the desirable narrow size distribution for all formulations.

Zeta potential (surface charge) is another important parameter that determines the stability of liposomal dispersions [70]. All the liposomes, demonstrated similar zeta potential values (Table 1), that is in the range of -9 mV to -14 mV, indicating relatively stable systems [71]. Thus, based on particle size and zeta potential, the developed liposomal formulations are theoretically stable that was confirmed experimentally by conducting the stability studies.

Finally, TEM analysis of liposomes shows spherical shaped particles with a single lipid bilayer (Figure 2), representing the expected morphology of unilamellar liposomal vesicles (ULV) [72,73]. The size of approximately 200 nm is consistent with the size range measured by dynamic light scattering (DLS) (Table 1). Some aggregated particles were observed in ultra-sonication and Mozafari method as shown in Figure 2.

### 5.2. Entrapment Efficiency (%EE)

The developed LC-MS/MS method (representative chromatogram shown in Figure 3) was able to separate and quantify four phytosterols (brassicasterol, campesterol, stigmasterol and β-sitosterol) and three tocopherols (alpha, gamma and delta). Both ultracentrifugation parameters and entrapment efficiencies were determined by analyzing bioactive using LC-MS/MS. The separation of the liposomes during ultracentrifugation was time-dependent. Relatively low amounts of liposomes sedimented after 30 min (around 80% for all bioactives) of ultracentrifugation, whereas high sedimentation of liposomes was observed at 60, 90 and 120 min. There was no significant difference in sedimentation at 60, 90 and 120 min of ultracentrifugation. This supports the notion that after 60 min of ultracentrifugation at RPM 32,000 (G-force of 138000), a significant amount of liposomes was sedimented, leaving free bioactives in the supernatant.

The optimum entrapment efficiencies of phytosterols and tocopherols into the liposomes obtained by the thin film hydration homogenization, thin film hydration ultra-sonication and Mozafari method is shown in Table 2. The results demonstrate that all three methods resulted in entrapment efficiency > 89% for phytosterols and tocopherols. Table 2 does not show any specific pattern in entrapment efficiency for bioactives. For example, in case of thin film hydration homogenization method, brassicasterol showed the highest entrapment efficiency among all phytosterols; however, in the case of the Mozafari method, brassicasterol has the lowest entrapment efficiency. Similarly, the Mozafari method showed the highest entrapment efficiency for gamma tocopherols among all tocopherols. On the other hand, thin film hydration ultrasonication showed the lowest entrapment efficiency for gamma tocopherol. Thus, no concrete conclusion was obtained regarding entrapment differences between these bioactives. The entrapment efficiency of some of lipophilic compounds were reported to be almost close to 100% [74,75]. However, Table 2 shows EE in the range of 89–97% for various bioactives, evaluated in our work. It is possible that some of the liposomes were too small and failed to sediment during ultracentrifugation. This will lead to decreased EE (the amount of bioactives in the sediment were taken as a basis to calculate EE). Nevertheless, the obtained EE (shown in Table 2) is consistent with entrapment efficiency of nutraceuticals such as vitamin E, resveratrol and retinol specified in the literature [57,65,76]. High entrapment efficiency, that is, greater than 85% is considered economical for industrial application because it eliminates the cost of separating free and entrapped bioactives that will be required in case of low entrapment efficiency.

### 5.3. Effect of Lyophilization on the Physicochemical Properties

Freeze-drying of liposomes resulted in the increment in particle size, reaching up to 500 nm (Figure 4). Various food compatible cryoprotectants such as sucrose, mannitol and lactose can be used to address this issue [77]. Thus, food compatible sucrose was tested as a cryoprotectant [56]. The addition of sucrose maintained the desired particle size (Figure 4). The lyophilized liposomes were then incorporated into model orange juice. Lyophilization is one of the crucial steps used for stabilization of liposomes [78]. It extends the shelf life of liposomes and can prevent thermosensitive bioactives from degradation [78].

### 5.4. Chemical Stability Studies

Pasteurization technique did not compromise the stability of bioactives as shown in Table 3. There was no significant change in the LC-MS/MS response for pasteurized and non-pasteurized formulations, ranging from 0.5 to 2.59% (Table 3). This shows that exposure to temperature of 72 °C for short time of 15 s does not degrade bioactives entrapped within liposomes in the model juice

### 5.5. Physical Stability Studies

Both pasteurized and non-pasteurized liposomes in model orange juice showed similar trend in particle size (Figure 5). This implies that high temperature in HTST pasteurization process did not compromise the stability of the liposomes. Further, particle size of vesicle did not change significantly during the one-month storage at 4 °C (Figure 5). This shows that liposomal orange juice can be stored in 4 °C for a month with adequate stability. Regarding zeta potential, unlike liposomes in purified water, liposomal model juice was found to have positive zeta potential in the range of 5.6–8.9 mV. Even though this zeta potential value is generally considered an indicator of instability to the colloidal system [71], liposomal model orange juice showed adequate storage stability. Probably, the optimized smaller vesicular size maintained the stability of particles preventing its aggregation.

## 6. Conclusions

To address the lipophilicity, heat and light sensitivity challenges, unilamellar liposomes containing phytosterols obtained from CODD and tocopherols were formulated and were applied to develop a functional juice. Three different formulation approaches were employed and were compared for their suitability in formulating phytosterols and tocopherols. All three methods showed optimum physicochemical properties and excellent entrapment efficiencies that were greater than 89%. Mozafari method was found to be simple and quick for formulating liposomes; however, the use of high temperature can possibly degrade thermosensitive bioactives. In addition, its low B/L ratio (not economical for scaling up) makes the Mozafari method less suitable method for phytosterols and tocopherols in comparison to thin film hydration ultrasonication and thin film hydration homogenization method. Both ultrasonication and homogenization seemed to be equally suitable at a laboratory scale. At an industrial scale, however, the homogenization method is more feasible due to the availability of homogenizers of large capacity. Thus, thin film hydration-homogenization seems to be the best method for scaling-up the liposomal formulation containing phytosterols and tocopherols. The pasteurization technique did not affect the chemical stability of tested bioactives. Moreover, model orange juice containing liposomes maintained an adequate physical stability during a period of one-month storage at 4 °C. In the future, liposomes containing phytosterols will be tested for cholesterol-lowering efficacy by conducting animal and human trials.

## Figures and Tables

**Figure 1 pharmaceutics-11-00185-f001:**
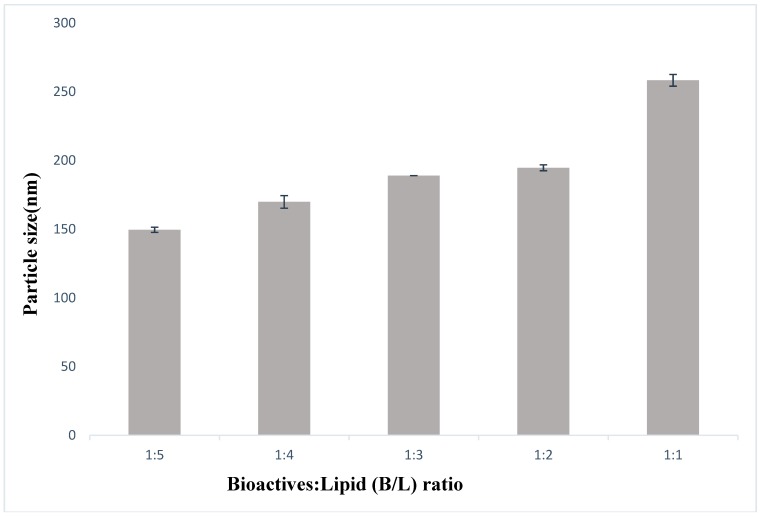
Particle size of liposomes prepared at different B/L ratio by homogenization method expressed as mean ± standard deviation.

**Figure 2 pharmaceutics-11-00185-f002:**
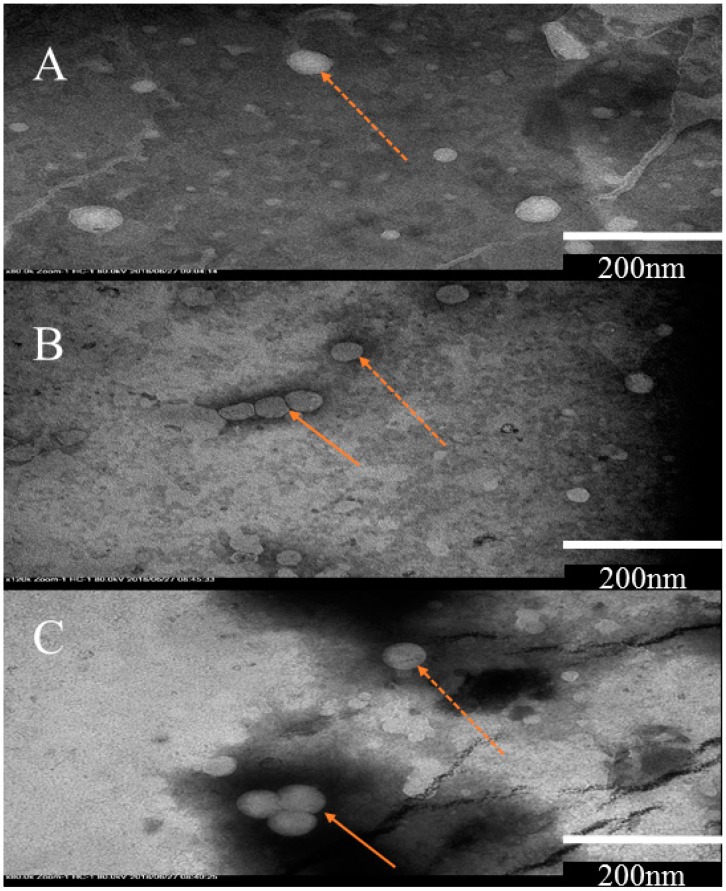
Transmission electron microscopy (TEM) analysis of liposomes prepared by; (**A**) homogenization method, (**B**) Ultrasonication and (**C**) Mozafari method. Sample of unilamellar vesicles are shown with a dotted arrow while aggregates are indicated by solid arrows. Scale bar in the figure A, B and C indicates 200nm, which represents the size of vesicle.

**Figure 3 pharmaceutics-11-00185-f003:**
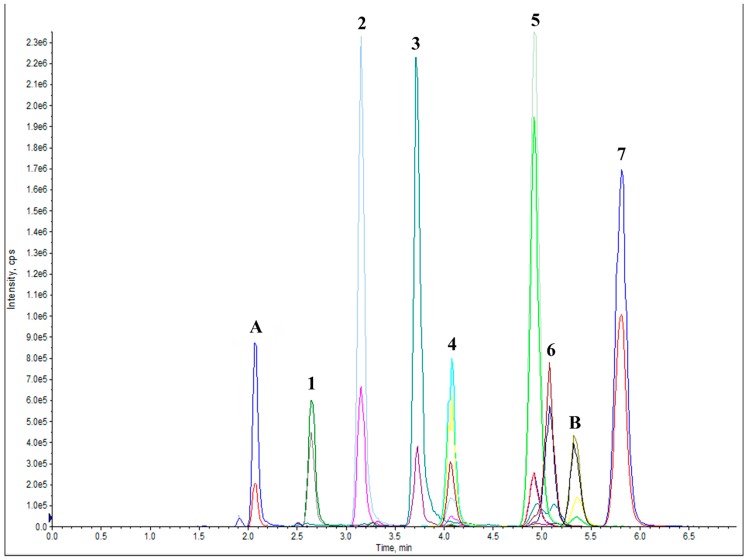
LC-MS/MS chromatogram of tocopherols: 1-δ tocopherol, 2-γ tocopherol, 3-α tocopherol; and phytosterols: 4-brassicasterol, 5-campesterol, 6-stigmasterol and 7-β-sitosterol. A-Rac tocol and B-cholestanol are internal standard.

**Figure 4 pharmaceutics-11-00185-f004:**
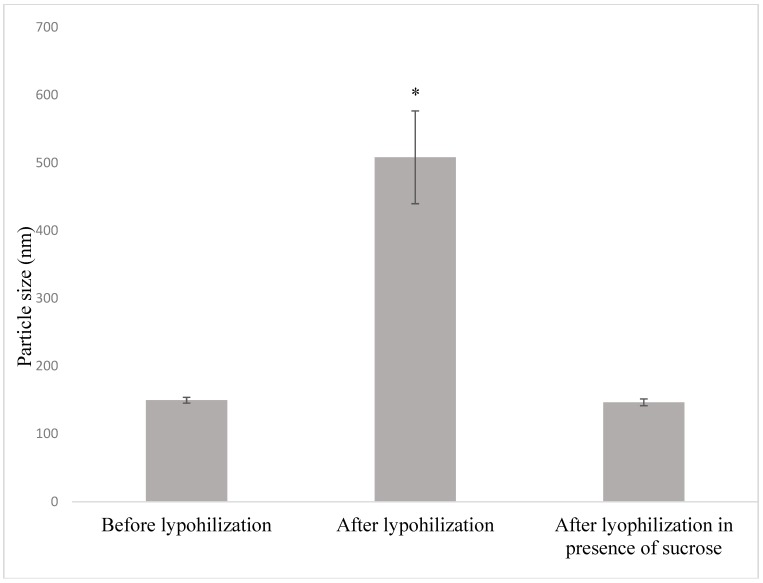
Effect of addition of sucrose as a cryo-protectant on particle size of liposomes before and after lyophilization expressed as mean ± standard deviation where * represents statistical significant (**p* < 0.05) in particle size after lyophilization in comparison to before lyophilization and after lyophilization in presence of sucrose.

**Figure 5 pharmaceutics-11-00185-f005:**
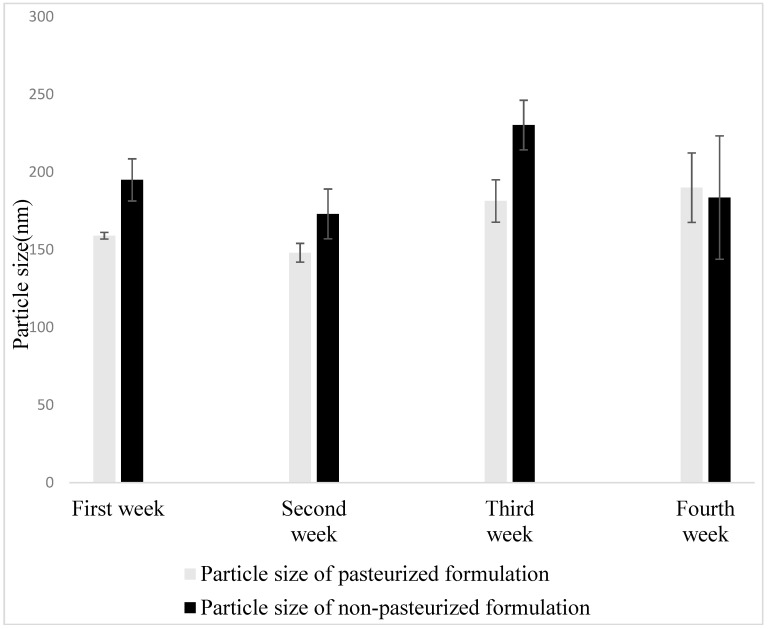
Particle size of pasteurized and non-pasteurized liposomes incorporated in model orange juice during storage period of 1 month at 4 °C expressed as mean ± standard deviation.

**Table 1 pharmaceutics-11-00185-t001:** Average particle size (nm), polydispersibility index and zeta potential (mv) of liposomes prepared by thin film hydration homogenization, thin film hydration ultrasonication and Mozafari method expressed as mean ± standard deviation where * represents statistical significant (**p* < 0.05) in average particle size of Mozafari method in comparison with homogenization and ultrasonication method.

Formulation Techniques	Average Particle Size (nm)	Polydispersibility Index (PDI)	Zeta Potential (mV)
Thin film hydration Homogenization	186.3 ± 4.4	0.370 ± 0.001	−13.0 ± 5.0
Thin film hydration ultra-sonication	196.2 ± 16.1	0.294 ± 0.084	−14.0 ± 3.4
Mozafari method	260.0 ± 23.0^*^	0.348 ± 0.087	−9.8 ± 0.3

**Table 2 pharmaceutics-11-00185-t002:** Entrapment efficiency of bioactives (phytosterols and tocopherols) into liposomes prepared by the thin film hydration homogenization, thin film hydration ultra-sonication and Mozafari method expressed as mean ± standard deviation.

Methods	Entrapment Efficiency (EE %)					
	Brassicasterol	Campesterol	β-Sitosterol	Alpha Tocopherol	Gamma Tocopherol	Delta Tocopherol
Thin film hydration-Homogenization	95.9 ± 1.7	94.0 ± 2.2	94.8 ± 3.0	91.6 ± 2.4	90.5 ± 2.9	91.6 ± 3.6
Thin film hydration-Ultrasonication	91.5 ± 2.4	92.3 ± 3.4	90.1 ± 1.9	91.2 ± 2.1	89.8 ± 3.1	90.1 ± 2.3
Mozafari method	89.4 ± 2.8	93.7 ± 6.0	93.1 ± 6.0	92.3 ± 7.5	97.4 ± 1.9	95.3 ± 1.4

**Table 3 pharmaceutics-11-00185-t003:** Relative change in the concentration (represented by area under curve, AUC) of phytosterols and tocopherol before and after pasteurization.

Bioactives	AUC of Non-Pasteurized Bioactives	AUC of Pasteurized Bioactives	Percentage Relative Change in AUC (%) of Pasteurized and Non-Pasteurized
Brassicasterol	5.63 × 10^6^	5.60 × 10^6^	0.53
Campesterol	2.32 × 10^7^	2.26 × 10^7^	2.59
β-sitosterol	3.40 × 10^6^	3.37 × 10^6^	0.88
α-tocopherol	2.76 × 10^7^	2.72 × 10^7^	1.45
γ-tocopherol	4.94 × 10^6^	4.89 × 10^6^	1.01
δ-tocopherol	4.84 × 10^6^	4.73 × 10^6^	2.27

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
