# Peer review of "Development and Characterization of Liposomal Formulations Containing Phytosterols Extracted from Canola Oil Deodorizer Distillate along with Tocopherols as Food Additives"

_pharmaceutics, 2019, doi:10.3390/pharmaceutics11040185_

Round 1

Reviewer 1 Report

This paper describes the characterization of a lipid formulation of a potentially significant group of nutraceuticals. The authors hope ultimately to study the effects of these formulations in vivo. The work is thoroughly carried out and complete. I have one suggestion for the authors. The compounds being incorporated here into a phospholipid suspension are very lipophilic, and I would not generally think of them as substances that would be appreciably soluble in water. What, then, are the authors saying when they observe less than 100% incorporation into liposomes? Given the methodology being used, it is possible that the result is simply material lost in processing. It is also possible that there is indeed a residue that is not isolated by sedimentation, but that it is in the form of a subfraction of the liposomes that are too small to sediment. Since the authors have not attempted to measure what is in the supernatant, and have not also measured the phospholipid content, it is difficult to distinguish these possibilities, and also whether there might actually be a residue of the sterols and tocopherols in solution in the supernatant. I don't think it important that this be resolved at this stage, but a brief discussion of these possibilities would strengthen the paper.

Author Response

Reviewer 1

This paper describes the characterization of a lipid formulation of a potentially significant group of nutraceuticals. The authors hope ultimately to study the effects of these formulations in vivo. The work is thoroughly carried out and complete. I have one suggestion for the authors. The compounds being incorporated here into a phospholipid suspension are very lipophilic, and I would not generally think of them as substances that would be appreciably soluble in water. What, then, are the authors saying when they observe less than 100% incorporation into liposomes? Given the methodology being used, it is possible that the result is simply material lost in processing. It is also possible that there is indeed a residue that is not isolated by sedimentation, but that it is in the form of a subfraction of the liposomes that are too small to sediment. Since the authors have not attempted to measure what is in the supernatant, and have not also measured the phospholipid content, it is difficult to distinguish these possibilities, and also whether there might actually be a residue of the sterols and tocopherols in solution in the supernatant. I don't think it important that this be resolved at this stage, but a brief discussion of these possibilities would strengthen the paper.

Response

We thank the reviewer and we agree with their point of view. In fact, we attempted to measure the target analytes within the supernatant, but the concentration was below the LLOQ. To clarify, the following has been added in the discussions:

Sections 5.2 “The entrapment efficiency of some of lipophilic compounds were reported to be almost close to 100% [74,75]. However, Table 2  shows EE in the range of 89-97% for various bioactives, evaluated in our work. It is possible that some of the liposomes were too small and failed to sediment during ultracentrifugation. This will lead to decreased EE (the amount of bioactives in the sediment were taken as a basis to calculate EE). Nevertheless, the obtained EE (shown in Table 2) is consistent with entrapment efficiency of nutraceuticals such as vitamin E, resveratrol and retinol specified in the literature [57,65,76]”.

Reviewer 2 Report

Dear Author

Your paper “Development and characterization of lipid-based formulations containing phytosterols extracted from Canola oil deodorizer distillate along with tocopherols as food additivesis suitable for publication in the Pharmaceutics. The subject has good scientific value, but some aspects need to be improved. Therefore, I suggest to revise the manuscript, according to my comments.

1) The authors developed PC liposomes in their study. I recommend changing the title from “lipid-based formulations” to liposomes.

2) Pag 2 lines 61-63: the authors reported “All of these shortcomings can be addressed by employing lipid-based formulations that cost low to scale up, require low heat and low quantities of surfactants or emulsifiers”. What formulations do they refer to? The scale up of liposomes presents difficulties. In the case of NLC and SLN or micro nanoemulsions the amount of surfactant is often high. Authors must reconsider the sentence.

3) Pag 3 line 95. In the work the authors developed and characterized the formulation, and performed a stability study. They did not evaluate efficacy with in vivo studies.

4) In the experimental section: combine paragraph 3.4 with 3.6. The procedure to define EE% is not clear.

5) Paragraph: 3.1.3:  Are phytosterols and tocopherol mixture stable at 110°C? How did the authors evaluate it?

6) The validation of the methods and the validation parameters values are missing in the Results. A chromatographic profile must be reported.

7) The authors evaluated only the physical stability. I believe it is important also to evaluate the chemical stability. How was the sample prepared after the pasteurization process of the model juice?

8) Paragraph: 5.1: Why with the Montafari method is a higher ratio bioactives:lipides necessary? Explain better in the results.

9) Have the authors tested other cryoprotectants?

10) In the conclusions the authors must highlight the best method of preparing liposomes, even considering an industrial application.

Author Response

Your paper “Development and characterization of lipid-based formulations containing phytosterols extracted from Canola oil deodorizer distillate along with tocopherols as food additives” is suitable for publication in the Pharmaceutics. The 

Below are the responses to the constructive comments of the reviewer and I attached a document as well that includes both these comments as well as, an equation, figure and table that I could not just insert in this box. 

subject has good scientific value, but some aspects need to be improved. Therefore, I suggest to revise the manuscript, according to my comments.

 1) The authors developed PC liposomes in their study. I recommend changing the title from “lipid-based formulations” to liposomes.

Response: We agree with the reviewer, title has been changed from “lipid based formulations” to “liposomal formulations” as shown below.

“Development and characterization of liposomal formulations containing phytosterols extracted from Canola oil deodorizer distillate along with tocopherols as food additives”

2) Pag 2 lines 61-63: the authors reported “All of these shortcomings can be addressed by employing lipid-based formulations that cost low to scale up, require low heat and low quantities of surfactants or emulsifiers”. What formulations do they refer to? The scale up of liposomes presents difficulties. In the case of NLC and SLN or micro nanoemulsions the amount of surfactant is often high. Authors must reconsider the sentence.

Response: We thank the reviewer for their comment, and we agree- clarity is needed, since our intent is liposomes in particular and not all lipid-based formulations. The lines are revised as shown below:

“Thus, the selection of suitable formulation approach is crucial during the development of functional food that contain these bioactives. Encapsulation techniques, such as spray drying, fluidized bed coating, microemulsification and liposomal entrapment are emerging in the food industry to address lipophilicity related challenges [22,23]. Unfortunately, most of these techniques have shortcomings such as usage of high temperature (can possibly degrade phytosterols and tocopherols) and the requirement of large quantities of emulsifiers and surfactants, which are deleterious to human health [23-27]. All of these shortcomings can be addressed  by employing  liposomal formulations that  require low heat and low quantities of surfactants or emulsifiers [28]”.

To Shed additional light about the scale-up possibility, the following is added to the conclusions:

“Both ultrasonication and homogenization seemed to be equally suitable at a laboratory scale. At an industrial scale, however, the homogenization method is more feasible due to the availability of homogenizers of large capacity. Thus, thin film hydration-homogenization seems to be the best method for scaling-up the liposomal formulation containing phytosterols and tocopherols”.

3) Pag 3 line 95. In the work the authors developed and characterized the formulation, and performed a stability study. They did not evaluate efficacy with in vivo studies.

Response: The study is focused on product development for industrial application in this manuscript. We fully agree with the reviewer that in vivo evaluation is needed. In fact, we will be conducting in vivo studies in the close future, which will be reported upon completion. We have highlighted in vivo studies as a future work in conclusions as follows:

“In the future, liposomes containing phytosterols will be tested for cholesterol lowering efficacy by conducting animal and human trials.”

4) In the experimental section: combine paragraph 3.4 with 3.6. The procedure to define EE% is not clear.

Response: We thank the reviewer. Paragraph 3.4 and 3.6 have been combined. EE% has been defined as follows:

“3.5 Entrapment efficiency(EE)

In order to determine entrapment efficiency, free and entrapped bioactives were separated using ultracentrifugation. Beckman coulter ultracentrifuge (Beckman coulter, Inc, Indianapolis, United states) with rotor SW 60Ti was used for ultracentrifugation. Briefly, 5mL of liposomes was ultracentrifuged at 30, 60, 90 and 120 min at constant RPM 32,000 (G-force of 138000). The sediment at each time were analyzed using a validated LC-MS/MS method to optimize ultracentrifugation parameters. The liposomes (present in sediment) separated by ultracentrifugation were lyophilized using freeze dryer for 24 hours. Similar lyophilization process was employed with 5mL of unseparated liposomes for 24 hours. Dried unseparated and separated liposomes were dissolved in 2mL of chloroform separately. Aliquot of each were spiked with internal standard and diluted with acetonitrile. Samples were injected in LC-MS along with freshly prepared calibration and quality control standards, as described [54].The entrapment efficiency  was calculated by measuring the ratio of entrapped bioactives in the formulation to the total bioactives present in the formulation and was determined  using following equation:

Where,

E bioactives = Entrapped bioactives in liposomes (present in  sediment of separated liposomes)

T bioactives = Total bioactives in liposomes (present in unseparated liposomes)                                  

5) Paragraph: 3.1.3:  Are phytosterols and tocopherol mixture stable at 110°C? How did the authors evaluate it?

Response: We thank the reviewer for the insightful comment. We believe that the reviewer is referring to the mozafari method. Yes, phytosterols and tocopherols are thermosensitive compounds. They can possibly undergo oxidation at high temperatures. Thus, mozafari method, which uses high temperature, is indeed not suitable for formulating phytosterols and tocopherols. Thus, the following sentence has been added in the conclusion:

“Mozafari method was found to be simple and quick for formulating liposomes; however, the use of high temperature can possibly degrade thermosensitive biaoctives. In addition, its low B/L ratio (not economical for scaling up) makes the mozafari method less suitable method for phytosterols and tocopherols in comparison to thin film hydration ultrasonication and thin film hydration homogenization method”.

 6) The validation of the methods and the validation parameters values are missing in the results. A chromatographic profile must be reported.

Response: The method has been fully developed and validated and we are currently preparing a detailed method development and validation paper.  We, however, provided description of the method and cited a conference poster.

As per the reviewer’s recommendation, chromatographic profile has been added to the paper in Figure 3.

"Figure 3: LC-MS/MS chromatogram of tocopherols: 1-δ tocopherol, 2-γ tocopherol, 3-α tocopherol; and phytosterols: 4-brassicasterol, 5-campesterol, 6-stigmasterol and 7-β-sitosterol. A-Rac tocol and B-cholestanol are internal standard".

7) The authors evaluated only the physical stability. I believe it is important also to evaluate the chemical stability. How was the sample prepared after the pasteurization process of the model juice?

Response: We agree with the reviewer and this is indeed important. To address this, we conducted additional chemical stability evaluation using LC-MS/MS. There was no change in concentration of biaoctives in non-pasteurized and pasteurized formulation (relative quantification), showing that pasteurization did not compromise the stability of the bioactives. Section 3.8 has been added in the method section and section 5.4 is added in the result section as follows:

“Methods

“3.8 Chemical stability studies 

Both pasteurized and non-pasteurized model juice containing liposomal bioactives were analyzed using LC-MS/MS to assess the degradation of bioactive upon exposure to pasteurization temperature. Briefly, 5ml each of pasteurized and non-pasteurized liposomal model juice were lyophilized. Dried sample were dissolved in chloroform and were diluted with acetonitrile for LC-MS/MS analysis, as described. The LC-MS/MS response was compared to obtain relative quantification data.

Results

5.4. Chemical stability studies

Pasteurization technique did not compromise the stability of biactives as shown in Table 3. There was no significant change in the LC-MS/MS response for pasteurized and non-pasteurized formulations, ranging 0.5-2.59% (Table 3). This shows that exposure to temperature of 72°C for short time of 15 seconds does not degrade bioactives entrapped within liposomes in the model juice

Table 3: Relative change in the concentration (represented by area under curve, AUC) of phytosterols and tocopherol before and after pasteurization

Regarding How was the sample prepared after the pasteurization process of the model juice?

Response: We assume that reviewer is referring to the sample preparation for stability studies. Freeze dried liposomes is added to a model juice by vortexing for 5mins and finally liposomal model juice was pasteurized. After the pasteurization of model juice containing liposomal bioactives, the sample was stored in 4°C freezer and every week physical stability of the liposomal model juice was assessed using zeta sizer.  For chemical stability evaluation, freeze-dried liposomal model juice (both pasteurized and non-pasteurized) were dissolved in chloroform and was diluted with acetonitrile for LC-MS analysis, as described above. The following is added regarding the physical stability:

“3.9 Physical stability studies

Physical stability evaluation was conducted at the interval of 7 days for a month. Particle size of pasteurized and non-pasteurized liposomes incorporated both in model orange juice were analyzed.”

 8) Paragraph: 5.1: Why with the Montafari method is a higher ratio bioactives:lipides necessary? Explain better in the results

Response: We thank the reviewer. The following has been added in the result to explain why the mozafari method can possibly require high lipids making bioactives:lipids(B/L) ratio of 1:5.

“Loading techniques along with the preparation procedures are found to influence drug/lipid ratio of liposomes[67]. In both thin film hydration homogenization and thin film hydration ultrasonication, hydrated bioactives-PC film is subjected to cavitation and shearing forces unlike the mozafari method where less intense magnetic stirring is used during the loading process. This might have led to differences in the B/L ratio of the mozafari method in comparison with ultrasonication and homogenization methods”.

9) Have the authors tested other cryoprotectants?

Response: With the aim of using food compatible cryo-protectant, sucrose was tested first. Usage of 5%w/v sucrose maintained the initial size (before lyophilization) of liposomes. Thus, other cryo-protectant were not evaluated.

10) In the conclusions the authors must highlight the best method of preparing liposomes, even considering an industrial application

Response: We agree with the reviewer; we now highlighted the best method for preparing phytosterols and tocopherols containing liposomes, even considering an industrial application in the conclusions:

Conclusion

“Mozafari method was found to be simple and quick for formulating liposomes; however, the use of high temperature can possibily degrade thermosensitive bioactives. In addition, its low B/L ratio (not economical for scaling up) makes the mozafari method less suitable method for phytosterols and tocopherols in comparison to thin film hydration ultrasonication and thin film hydration homogenization method. Both ultrasonication and homogenization seemed to be equally suitable at a laboratory scale. At an industrial scale, however, the homogenization method is more feasible due to the availability of homogenizers of large capacity. Thus, thin film hydration-homogenization seems to be the best method for scaling-up the liposomal formulation containing phytosterols and tocopherols. The pasteurization technique did not affect the chemical stability of tested bioactives. Moreover, model orange juice containing liposomes maintained an adequate physical stability during a period of one-month storage at 4°C. In the future, liposomes containing phytosterols will be tested for cholesterol lowering efficacy by conducting animal and human trials”.

Round 2

Reviewer 2 Report

The authors replied to my reviews in a comprehensive manner and they implemented the quality of the manuscript. The paper is acceptet in this form.